# Is Contrastive Learning Necessary? A Study of Data Augmentation vs Contrastive Learning in Sequential Recommendation

## ABSTRACT

Sequential recommender systems (SRS) are designed to predict users' future behaviors based on their historical interaction data. Recent research has increasingly utilized contrastive learning (CL) to leverage unsupervised signals to alleviate the data sparsity issue in SRS. In general, CL-based SRS first augments the raw sequential interaction data by using data augmentation strategies and employs a contrastive training scheme to enforce the representations of those sequences from the same raw interaction data to be similar. Despite the growing popularity of CL, data augmentation, as a basic component of CL, has not received sufficient attention. This raises the question: *Is data augmentation sufficient to achieve superior recommendation results?* To answer this question, we benchmark a large amount of data augmentation strategies, as well as state-of-the-art CL-based SRS methods, on four real-world datasets under both warm- and cold-start settings. Intriguingly, the conclusion drawn from our study is that *data augmentation is sufficient and CL may not be necessarily required*. In fact, utilizing augmentation alone can significantly alleviate the data sparsity issue and certain data augmentation can achieve similar or even superior performance compared with CL-based methods. We hope that our study can further inspire more fundamental studies on the key functional components of complex CL techniques. Our processed datasets and codes will be released once our paper is accepted.

## CCS CONCEPTS

• **Information systems → Recommender systems**.

## KEYWORDS

Data Augmentation, Sequential Recommendation, Contrastive Learning

**ACM Reference Format:**
Anonymous Author(s). 2023. Is Contrastive Learning Necessary? A Study of Data Augmentation vs Contrastive Learning in Sequential Recommendation. In *Proceedings of Make sure to enter the correct conference title from your rights confirmation emai (Conference acronym 'XX).* ACM, New York, NY, USA, 10 pages. https://doi.org/10.1145/nnnnnnn.nnnnnnn

## 1 INTRODUCTION

Sequential recommender systems (SRS) play a crucial role in various domains, such as e-commerce [1, 3, 41], video [7, 20], music [8, 26] and social media [12, 15]. The goal of these SRS is to predict the next item that a user is likely to interact with based on his/her historical behavior. One predominant obstacle in developing SRS is the data sparsity issue, where user-item interaction data is typically limited compared with a large number of users and items,

*Conference acronym 'XX, June 03–05, 2018, Woodstock, NY*
2023. ACM ISBN 978-1-4503-XXXX-X/18/06...$15.00
https://doi.org/10.1145/nnnnnnn.nnnnnnn

**Figure 1: (a) Direct data augmentation for sequential recommendation; (b) Contrastive learning for sequential recommendation.**

leading to insufficient training signals to learn informative item representations for the downstream recommendations.

Recently, contrastive learning for recommendation has attracted increasing attention due to its remarkable capability to enhance item representations by extracting self-supervised signals from raw user-item interaction data. Consequently, various contrastive learning-based SRS, such as CL4SRec [38], CoSeRec [22], ICLRec [4] and DuoRec [28], have been proposed. The core idea of these methods can be summarized into two interrelated steps: (1) generating positive views and negative views through data augmentation strategies; and (2) minimizing (resp. maximizing) the distance between positive (resp. negative) views using a contrastive loss function (such as InfoNCE [25]). As shown in Fig. 1, in these methods, data augmentation strategies are applied solely to the auxiliary tasks designed for contrastive learning rather than directly applied to the recommendation task itself. Such practice naturally raises two critical questions: *How is the recommendation performance when only data augmentation strategies are directly applied to the recommendation task? Can the performance of SRS be improved by solely relying on data augmentation (i.e., the first step) without using contrastive learning paradigm (i.e., the second step)?*

To answer these questions is crucial for revisiting the role of data augmentation strategies in sequential recommendation tasks. However, so far, the direct application of data augmentation to mitigate the data sparsity issue in sequential recommendation has not received sufficient attention. Only one study [30] has explored the effects of four augmentation strategies on sequential recommendation. Nevertheless, it still suffers from following limitations: First, the augmentation strategies compared are not comprehensive. For instance, strategies like reorder and delete, commonly used in contrastive learning for sequential recommendation (SR), can also be employed as standalone data augmentation methods. Second, comparison with contrastive learning methods are not conducted, which is essential for revisiting the effectiveness of contrastive learning in the sequential recommendation research. Third,

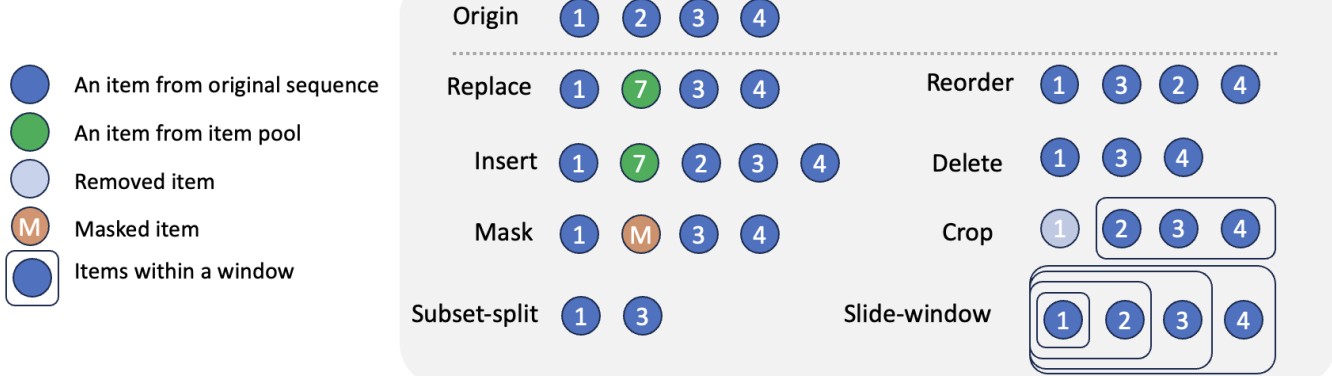

**Figure 2: Eight widely used sequence-level data augmentation strategies.**

insights or analysis regarding the factors that led to varying performance results of different augmentation strategies are not provided. Therefore, there is an urgent need to conduct a more systematic empirical study to thoroughly investigate the effectiveness of data augmentation in improving the performance of SRS.

To bridge this research gap, we conduct a comprehensive experimental study to compare the performance of SRS based on data augmentation only and that of SRS based on full contrastive learning. To be specific, we focus on investigating the effectiveness of eight popular sequence-level augmentation strategies: *insert, replace, crop, delete, mask, reorder, subset-split, and slide-window*. These augmentation strategies have been widely adopted in contrastive learning-based SRS over the past five years. Specifically, we decouple these sequence-level data augmentation strategies from contrastive learning methods and directly apply them to augment the training sequences. Both the original sequences and the augmented sequences are input together to backbone models (such as SASRec [16]) for training. Afterward, we benchmark these eight sequence-level data augmentation strategies and three state-of-the-art contrastive learning-based SRS on four widely used datasets. We also simulate different cold-start scenarios to further evaluate the applicability of sequence-level data augmentation. Furthermore, we conduct in-depth analysis on the impact of the size of data augmentations, as well as the computational efficiency of various augmentation strategies and contrastive learning methods.

The experimental results demonstrate that, when using SASRec as the backbone, certain sequence-level augmentation strategies can achieve comparative or even superior performance compared to contrastive learning-based methods, while requiring less training and inference time. This finding not only validates the feasibility of directly using sequence-level augmentation to alleviate data sparsity issue, but also suggests that the current research community might underestimate the effectiveness of simple sequence-level augmentation and overly emphasize the necessity of contrastive learning in sequential recommendation tasks.

The contribution of this paper can be summarized as follows:

- We decouple sequence-level augmentation strategies from contrastive learning methods and to benchmark these strategies in the context of sequential recommendation tasks.
- We explore the synergistic effects between slide-window strategy and other data augmentation techniques. We also verify that CL-based methods can benefit from slide-window strategy.
- Our experimental results demonstrate that employing specific sequence-level augmentation strategies can effectively mitigate the problem of data sparsity in sequential recommender systems. Moreover, these strategies require less training and inference time compared to CL-based methods.

## 2 PROBLEM FORMULATION

Let $\mathcal{U}, \mathcal{I}$ denote the sets of users and items, respectively. For a user $u \in \mathcal{U}$, the historical interactions of this user can be represented as $\mathcal{S}^u = [v_1^u, v_2^u, \ldots, v_{|\mathcal{S}^u|}^u]$, where $v_i^u \in \mathcal{I}$ is the *i-th* interaction in the chronologically ordered sequence $\mathcal{S}^u$ and $|\mathcal{S}^u|$ denotes the sequence length. The set of users' actions can be represented as $\mathcal{S} = \{\mathcal{S}^1, \mathcal{S}^2, \ldots, \mathcal{S}^{|\mathcal{U}|}\}$, where $|\mathcal{U}|$ is the number of users. Given the historical interaction sequence $\mathcal{S}^u$ of user $u$, the goal of sequential recommendation is to predict the next item $v_{next} \in \mathcal{I}$ that user $u$ will interact with at the $(|\mathcal{S}^u| + 1)$-*th* time step, denoted as $p(v_{next} \mid \mathcal{S}^u)$.

## 3 SEQUENCE-LEVEL DATA AUGMENTATION STRATEGIES

Our experiments include multiple common rule-based sequence-level data augmentation strategies. All of them can be viewed as operators that create augmented sequences by performing certain transformations to the original sequence. Fig. 2 illustrates how to augment each interaction sequence using these operators.

### 3.1 Item Insert

The "insert" action begins by selecting an insertion position, followed by the insertion of a chosen item from the item pool, resulting in an augmented sequence. For the user's interaction sequence $\mathcal{S}^u$,

let $i$ be an insertion position, and $t$ be a chosen item from the item set $\mathcal{I}$. The augmented sequence $\mathcal{S}^{u'}$ can be represented as:

$$\mathcal{S}^{u'} = [v_1^u, \ldots, v_{i-1}^u, t, v_i^u, \ldots, v_{|\mathcal{S}^u|}^u]. \tag{1}$$

## 3.2 Item Delete

The "delete" action randomly selects an item from the sequence for removal, thereby generating an augmented sequence. For $\mathcal{S}^u$ and a randomly chosen item at position $k$, the augmented sequence $\mathcal{S}^{u'}$ is:

$$\mathcal{S}^{u'} = [\,]v_1^u, \ldots, v_{k-1}^u, v_{k+1}^u, \ldots, v_{|\mathcal{S}^u|}^u]. \tag{2}$$

## 3.3 Item Replace

The "replace" action begins by selecting the item in the sequence that will be replaced, followed by selecting an item from the item pool to substitute the chosen item, resulting in an augmented sequence. For $\mathcal{S}^u$, let $j$ be a position of the item to be replaced, and $t'$ be a chosen item from $\mathcal{I}$. The augmented sequence $\mathcal{S}^{u'}$ is:

$$\mathcal{S}^{u'} = [v_1^u, \ldots, v_{j-1}^u, t', v_{j+1}^u, \ldots, v_{|\mathcal{S}^u|}^u]. \tag{3}$$

## 3.4 Item Crop

The "crop" action initially selects a cutoff position, from which a continuous series of items with a specified length are extracted as an augmented sequence. Let $c$ be the cutoff position and $l$ be the length of the cropped sequence. The augmented sequence $\mathcal{S}^{u'}$ is:

$$\mathcal{S}^{u'} = [v_c^u, v_{c+1}^u, \ldots, v_{c+l-1}^u]. \tag{4}$$

## 3.5 Item Mask

The "mask" action initially chooses an item from the sequence and subsequently masks the ID of the selected item using a predefined mask symbol. Let $m$ be the position of the chosen item from sequence $\mathcal{S}^u$. If $\mu$ is the predefined mask symbol, the sequence after masking can be given as:

$$\mathcal{S}^{u'} = [v_1^u, \ldots, v_{m-1}^u, \mu, v_{m+1}^u, \ldots, v_{|\mathcal{S}^u|}^u]. \tag{5}$$

## 3.6 Item Reorder

The "reorder" action initially selects a sub-sequence of a specific length and subsequently shuffles the order of the items within that sub-sequence. The sub-sequence and the remaining parts of the original sequence are then combined according to their original order, resulting in an augmented sequence. Consider a sub-sequence of length $r$ starting at position $d$ from $\mathcal{S}^u$. Let $\text{shuffle}(x)$ denote a function that shuffles the order of the elements in $x$. The augmented sequence $\mathcal{S}^{u'}$ after shuffling this sub-sequence is:

$$\mathcal{S}^{u'} = [v_1^u, \ldots, v_{d-1}^u, \text{shuffle}(v_d^u, \ldots, v_{d+r-1}^u), v_{d+r}^u, \ldots, v_{|\mathcal{S}^u|}^u]. \tag{6}$$

## 3.7 Subset Split

Similar to the dropout mechanism [31], for subset split method, each item $v_i^u$ in the original sequence $\mathcal{S}^u$ will be included in $\mathcal{S}^{u'}$ with a probability of $1 - \theta$ and the probability of discarding is $\theta$:

$$v_i^{u'} = \begin{cases} v_i^u, & P = 1 - \theta \\ \text{discarded}, & P = \theta \end{cases} \tag{7}$$

### Table 1: Statistics of the datasets after preprocessing.

| Dataset | Beauty | Sports | ML-1m | Yelp |
|---|---|---|---|---|
| # Users | 22,363 | 35,598 | 6,041 | 30,499 |
| # Items | 12,101 | 18,357 | 3,407 | 20,068 |
| # Avg. Actions / User | 8.9 | 8.3 | 165.5 | 10.4 |
| # Avg. Actions / Item | 16.4 | 16.1 | 292.6 | 15.8 |
| # Actions | 198,502 | 296,337 | 999,611 | 317,182 |
| Sparsity | 99.93% | 99.95% | 95.15% | 99.95% |

Thus, the augmented sequence $\mathcal{S}^{u'}$ is essentially a "subset" of the original sequence $\mathcal{S}^u$, which can be mathematically represented as:

$$\mathcal{S}^{u'} = [v_1^{u'}, v_2^{u'}, \ldots, v_{|\mathcal{S}^{u'}|}^{u'}]. \tag{8}$$

Note that the length of the augmented sequence $|\mathcal{S}^{u'}|$ can vary due to the random discarding process and is likely to be less than or equal to $|\mathcal{S}^u|$.

## 3.8 Slide-window

The slide-window strategy employs a window of a designated length, denoted as $L$, to extract a cropped sequence of augmented items at each step. The window initiates sliding with its right edge positioned to the left of the first item, and the sliding process concludes when the right edge reaches the last item. For a given window length $L$, at each step $t$, the cropped sequence of augmented items from $\mathcal{S}^u$ is:

$$\mathcal{S}_t^{u'} = [v_t^u, v_{t+1}^u, \ldots, v_{t+L-1}^u], \tag{9}$$

where the sliding process iterating until $t + L - 1 = |\mathcal{S}^u|$.

The selection of augmentation positions and augmented items in the aforementioned data augmentation strategies is obtained through random sampling according to a uniform distribution. For the slide-window strategy, the size of augmentations depends on the original sequence length and window length. For other augmentation strategies, their size of augmentations is regarded as a hyperparameter $n$. We discuss the impact of the size of augmentations $n$ in Sec. 4.4.

## 4 EXPERIMENT

In this section, we conduct comprehensive experiments to answer the following key research questions:

- **RQ1:** How do different sequence-level augmentation strategies compare against state-of-the-art contrastive learning based SR methods?
- **RQ2:** How do sequence-level augmentation and contrastive learning methods perform in cold-start scenarios?
- **RQ3:** Why some direct data augmentation methods, such as slide-window, can achieve better performances compared with contrastive learning-based methods?
- **RQ4:** Does the size of augmentations, sampling strategy, slide-window size and model architecture affect the performance of sequence-level augmentation?
- **RQ5:** How does the training time and inference time of sequence-level data augmentation compare to contrastive learning methods?

**Table 2: Comparison of different data augmentation strategies and contrastive learning-based SR methods. Cells are blue if the augmentation strategy boosts SASRec performance, and orange if the augmentation hurts the performance.**

| Model | Beauty | | | | Sports | | | | Yelp | | | | ML-1m | | | |
|---|---|---|---|---|---|---|---|---|---|---|---|---|---|---|---|---|
| | Recall | | NDCG | | Recall | | NDCG | | Recall | | NDCG | | Recall | | NDCG | |
| | @10 | @20 | @10 | @20 | @10 | @20 | @10 | @20 | @10 | @20 | @10 | @20 | @10 | @20 | @10 | @20 |
| SASRec | 4.52±0.04 | 6.47±0.15 | 2.24±0.03 | 2.73±0.05 | 2.32±0.08 | 3.41±0.08 | 1.18±0.05 | 1.46±0.05 | 3.51±0.07 | 4.68±0.09 | 2.32±0.03 | 2.61±0.06 | 5.63±0.40 | 8.45±0.30 | 2.79±0.15 | 3.50±0.12 |
| + subset-split | 4.78±0.08 | 6.98±0.14 | 2.40±0.02 | 2.95±0.03 | 2.61±0.08 | 3.89±0.07 | 1.32±0.07 | 1.65±0.07 | 3.77±0.04 | 5.17±0.11 | 2.43±0.07 | 2.78±0.05 | 7.57±0.56 | 11.09±0.60 | 3.68±0.26 | 4.57±0.26 |
| + crop | 5.28±0.10 | 7.72±0.05 | 2.61±0.05 | 3.22±0.04 | 2.86±0.03 | 4.28±0.04 | 1.41±0.03 | 1.77±0.03 | 4.17±0.14 | 5.79±0.13 | 2.69±0.05 | 3.10±0.03 | 9.08±0.42 | 13.51±0.48 | 4.23±0.18 | 5.35±0.19 |
| + delete | 4.83±0.12 | 7.10±0.04 | 2.39±0.09 | 2.96±0.06 | 2.52±0.06 | 3.80±0.08 | 1.26±0.04 | 1.59±0.05 | 3.73±0.08 | 5.11±0.13 | 2.42±0.03 | 2.76±0.01 | 6.75±0.12 | 9.59±0.26 | 3.29±0.03 | 4.01±0.09 |
| + mask | 4.26±0.06 | 6.29±0.21 | 2.07±0.05 | 2.58±0.08 | 2.13±0.04 | 3.17±0.04 | 1.06±0.03 | 1.32±0.03 | 3.28±0.11 | 4.38±0.13 | 2.30±0.05 | 2.58±0.04 | 6.05±0.10 | 8.86±0.08 | 2.87±0.03 | 3.58±0.04 |
| + reorder | 4.67±0.08 | 6.87±0.08 | 2.28±0.06 | 2.83±0.06 | 2.45±0.05 | 3.67±0.08 | 1.22±0.01 | 1.53±0.02 | 3.60±0.07 | 4.95±0.10 | 2.42±0.07 | 2.76±0.05 | 5.84±0.19 | 8.54±0.33 | 2.8±0.08 | 3.47±0.10 |
| + insert | 4.62±0.12 | 6.79±0.11 | 2.28±0.05 | 2.83±0.05 | 2.38±0.09 | 3.54±0.13 | 1.17±0.07 | 1.46±0.08 | 3.62±0.05 | 4.87±0.05 | 2.46±0.12 | 2.78±0.11 | 6.67±0.54 | 9.76±0.27 | 3.23±0.24 | 4.01±0.16 |
| + replace | 4.26±0.09 | 6.20±0.09 | 2.08±0.05 | 2.57±0.04 | 2.05±0.07 | 3.07±0.09 | 1.01±0.03 | 1.26±0.04 | 3.23±0.06 | 4.22±0.05 | 2.33±0.04 | 2.57±0.04 | 5.90±0.15 | 8.64±0.16 | 2.79±0.05 | 3.48±0.04 |
| + slide-window | 7.81±0.05 | 11.41±0.19 | 3.82±0.06 | 4.74±0.09 | 5.00±0.08 | 7.43±0.15 | 2.30±0.03 | 2.92±0.05 | 5.85±0.05 | 8.47±0.09 | 3.73±0.04 | 4.39±0.04 | 19.78±0.32 | 28.9±0.30 | 10.37±0.23 | 12.67±0.20 |
| CL4SRec | 5.20±0.13 | 7.87±0.15 | 2.65±0.02 | 3.32±0.03 | 3.32±0.07 | 5.10±0.09 | 1.66±0.05 | 2.11±0.05 | 4.04±0.07 | 5.88±0.11 | 2.53±0.05 | 2.99±0.06 | 4.74±0.33 | 7.4±0.27 | 2.33±0.14 | 3±0.13 |
| CoSeRec | 4.72±0.06 | 7.01±0.07 | 2.31±0.04 | 2.89±0.04 | 2.71±0.06 | 4.12±0.03 | 1.33±0.03 | 1.68±0.01 | 4.03±0.07 | 5.53±0.10 | 2.65±0.03 | 3.03±0.03 | 5.83±0.21 | 8.47±0.29 | 2.82±0.09 | 3.49±0.11 |
| ICLRec | 4.84±0.05 | 7.14±0.12 | 2.42±0.05 | 3.01±0.06 | 2.66±0.03 | 4.00±0.09 | 1.30±0.03 | 1.64±0.03 | 3.46±0.04 | 4.53±0.06 | 2.37±0.02 | 2.64±0.03 | 5.87±0.1 | 8.74±0.2 | 2.78±0.03 | 3.5±0.08 |

**Table 3: Comparison of slide-window (SW) combined with different data augmentation strategies or contrastive learning methods. Cells are blue if the combination boosts SASRec performance, and orange if the combination hurts the performance.**

| Model | Beauty | | | | Sports | | | | Yelp | | | | ML-1m | | | |
|---|---|---|---|---|---|---|---|---|---|---|---|---|---|---|---|---|
| | Recall | | NDCG | | Recall | | NDCG | | Recall | | NDCG | | Recall | | NDCG | |
| | @10 | @20 | @10 | @20 | @10 | @20 | @10 | @20 | @10 | @20 | @10 | @20 | @10 | @20 | @10 | @20 |
| SASRec + SW | 7.81±0.05 | 11.41±0.19 | 3.82±0.06 | 4.74±0.09 | 5±0.08 | 7.43±0.15 | 2.30±0.03 | 2.92±0.05 | 5.85±0.05 | 8.47±0.09 | 3.73±0.04 | 4.39±0.04 | 19.78±0.32 | 28.9±0.30 | 10.37±0.23 | 12.67±0.20 |
| + subset-split | 8.27±0.18 | 12.05±0.23 | 4.12±0.11 | 5.07±0.12 | 5.43±0.10 | 8.11±0.06 | 2.53±0.04 | 3.20±0.04 | 6.18±0.05 | 9.08±0.06 | 3.85±0.02 | 4.58±0.03 | 19.87±0.18 | 28.59±0.29 | 10.38±0.13 | 12.57±0.16 |
| + crop | 7.29±0.17 | 10.5±0.11 | 3.70±0.10 | 4.51±0.10 | 4.55±0.09 | 6.84±0.11 | 2.12±0.03 | 2.70±0.03 | 5.23±0.07 | 7.38±0.08 | 3.44±0.04 | 3.98±0.03 | 17.36±0.30 | 25.58±0.25 | 8.97±0.18 | 11.04±0.17 |
| + delete | 8.22±0.18 | 11.97±0.15 | 4.08±0.11 | 5.02±0.09 | 5.19±0.14 | 7.84±0.21 | 2.38±0.06 | 3.05±0.08 | 5.98±0.14 | 8.75±0.06 | 3.75±0.06 | 4.44±0.03 | 20.33±0.17 | 29.39±0.27 | 10.57±0.13 | 12.86±0.14 |
| + mask | 7.47±0.11 | 10.96±0.11 | 3.65±0.06 | 4.52±0.06 | 4.4±0.07 | 6.78±0.12 | 2.01±0.04 | 2.61±0.05 | 5.51±0.12 | 8.04±0.14 | 3.53±0.06 | 4.17±0.07 | 20.22±0.20 | 29.32±0.12 | 10.50±0.20 | 12.80±0.19 |
| + reorder | 7.84±0.13 | 11.49±0.17 | 3.80±0.06 | 4.72±0.07 | 4.82±0.10 | 7.39±0.10 | 2.19±0.05 | 2.84±0.15 | 5.99±0.08 | 8.66±0.06 | 3.78±0.05 | 4.44±0.04 | 20.33±0.25 | 29.34±0.42 | 10.66±0.22 | 12.94±0.24 |
| + insert | 8.08±0.15 | 11.80±0.17 | 4.03±0.10 | 4.96±0.11 | 5.24±0.06 | 7.98±0.13 | 2.42±0.03 | 3.10±0.06 | 6.18±0.09 | 9.05±0.17 | 3.83±0.04 | 4.55±0.06 | 19.59±0.48 | 28.22±0.13 | 10.27±0.23 | 12.45±0.13 |
| + replace | 7.08±0.08 | 10.33±0.19 | 3.50±0.06 | 4.31±0.06 | 4.14±0.08 | 6.40±0.06 | 1.90±0.04 | 2.47±0.03 | 5.27±0.01 | 7.63±0.06 | 3.40±0.02 | 3.99±0.02 | 20.07±0.33 | 29.21±0.46 | 10.56±0.1 | 12.86±0.16 |
| CL4SRec + SW | 7.72±0.10 | 11.47±0.21 | 3.79±0.04 | 4.74±0.04 | 5.1±0.03 | 7.86±0.08 | 2.51±0.02 | 3.21±0.02 | 6.25±0.07 | 9.37±0.06 | 3.56±0.03 | 4.34±0.04 | 22.00±0.55 | 33.41±0.77 | 11.20±0.33 | 14.08±0.37 |
| CoSeRec + SW | 8.08±0.11 | 11.74±0.19 | 3.85±0.07 | 4.77±0.07 | 5.11±0.03 | 7.82±0.04 | 2.34±0.01 | 3.02±0.02 | 7.04±0.16 | 10.24±0.06 | 4.24±0.08 | 5.04±0.05 | 21.85±0.41 | 31.48±0.88 | 10.82±0.7 | 13.25±0.57 |
| ICLRec + SW | 8.11±0.09 | 11.75±0.06 | 3.82±0.04 | 4.73±0.04 | 5.34±0.10 | 8.09±0.16 | 2.42±0.04 | 3.12±0.05 | 6.52±0.07 | 9.46±0.09 | 3.98±0.05 | 4.72±0.05 | 22.68±0.36 | 31.74±0.37 | 12.24±0.32 | 14.53±0.33 |

## 4.1 Experimental Settings

*4.1.1 Dataset.* We conduct experiments on four widely used benchmark datasets with diverse distributions: **Beauty** and **Sports** is constructed from Amazon review datasets[1] [24]; **Yelp**[2] is a famous business recommendation dataset; **ML-1m**[3] is a famous movie rating dataset comprising 1 million ratings from 6,000 users on 4,000 movies. We pre-process these datasets in the same way following [16, 22, 34, 42, 43, 46] by removing items and users that occur less than 5 times. Tab. 1 shows dataset statistics after pre-processing.

*4.1.2 Evaluation Metrics.* Following previous works [4, 22, 32, 39], we use two widely adopted metrics to evaluate the performance of SR models: top-$K$ Recall (Recall@$K$) and top-$K$ Normalized Discounted Cumulative Gain (NDCG@$K$) with $K$ chosen from $\{10, 20\}$. For each user's interaction sequence, we reserve the last two items for validation and testing, respectively, and use the rest to train SR models. As suggested in [5, 18], we report the ranking results obtained over the whole item set for a fair comparison.

*4.1.3 Baseline Models.* The performance of eight data augmentation strategies is evaluated based on SASRec [16]. The details of these data augmentation strategies have been described in Sec 3. Additionally, three state-of-the-art contrastive learning-based sequential recommendation methods are chosen as baselines: (1) **CL4SRec** [38]: An invariant CL-based SR model that employs three sequence-level augmentation operators to generate positive pairs; (2) **CoSeRec** [22]: A SR model that introduces two informative

augmentation operators leveraging item correlations to create high-quality views for invariant contrastive learning; (3) **ICLRec** [4]: A general learning paradigm that leverages the clustered latent intent factor and contrastive self-supervised learning to optimize SR.

*4.1.4 Implementation Details.* To ensure a fair comparison, all baselines are implemented and evaluated using the popular recommendation framework RecBole [45] under identical settings. The models are trained with the Adam optimizer for 300 epochs, employing a batch size of 1024 and a learning rate of 0.001. For Beauty, Sports, and Yelp datasets, the maximum sequence length is set to 50, while for the ML-1m dataset, it is set to 200 due to its longer average sequence length. For attention-based methods, the dropout rate on the embedding matrix and attention matrix is set to 0.5 and we perform grid search on other hyper-parameters to find the best combination. The searching space is: number of self-attention layers $\in \{2, 3\}$, number of self-attention heads $\in \{2, 4\}$, hidden size $\in \{64, 128, 256\}$ and embedding $\in \{64, 128, 256\}$. Regarding data augmentation approaches, the slide-window has a length of 50 for Beauty, Sports, and Yelp, and 200 for ML-1m to accommodate the varying average sequence lengths [4]. Other hyperparameters for data augmentation are as follows: for "insert," "replace," "delete," and "mask," a single item is inserted, replaced, deleted, or masked; the dropout factor $\theta$ of subset split is set to 0.25; the length of subsequence in "crop" and "reorder" is set to 2. Both baselines and our method are carefully tuned on the used datasets for best performance. Additionally, we report the results based on experiments on 5 different random seeds. Mean ± std are reported in this paper.

---

[1] http://jmcauley.ucsd.edu/data/amazon/

[2] https://www.yelp.com/dataset

[3] https://grouplens.org/datasets/movielens/1m/

---

[4] Here we set the window length to be equal to the maximum sequence length, and leave the investigation of the window length's impact for future study.

## 4.2 Overall Performance (RQ1)

*4.2.1 Performance of single data augmentation strategy.* In this section, we explore the impact of eight sequence-level augmentation strategies on recommendation performance and compare them with three classic contrastive learning-based SR models. Specifically, for each sequence-level augmentation strategy, we select SASRec as the backbone. Each instance in the training set undergoes augmentation twice using the corresponding strategy. The results are presented in Tab. 2, and we draw the following observations:

**Most sequence-level data augmentations can improve the performance of SASRec.** Among the eight augmentation strategies, *slide-window* yields the best results, followed by *crop*. Specifically, *slide-window* achieves average relative performance improvements of **91.7%** and **80.6%** in terms of Recall@20 and NDCG@20, respectively, on datasets with shorter average sequence lengths (Beauty, Sports, and Yelp). Furthermore, on the ML-1m dataset with longer sequences, *slide-window* demonstrates more significant improvements, with Recall@20 and NDCG@20 increasing by **2.4x** and **2.6x**. Conversely, *mask* and *replace* perform poorly as data augmentation methods, reducing the performance of SASRec on Beauty, Sports, and Yelp datasets. This can be attributed to the detrimental impact of noise introduced by these methods on model training, particularly in shorter sequences.

**Some sequence-level data augmentation strategies outperform contrastive learning-based SR models.** Among them, *slide-window* performs better than all contrastive learning methods, while *cropping* achieves performance close to or even surpassing that of contrastive learning methods in most cases. It is worth noting that, on the ML-1m dataset with longer sequence lengths, all sequence-level data augmentation strategies can achieve performance comparable to or even superior to contrastive learning-based methods.

*4.2.2 Performance of combined data augmentation strategy.* Furthermore, we evaluate the performance of combining *slide-window* with other sequence-level augmentation strategies or contrastive learning methods, and summarize the results in Tab. 3. We observe that, in most cases, the performance of *slide-window + crop/mask /replace* is inferior to that of using *slide-window* alone. However, the combination of *slide-window* with the other four augmentation strategies, namely *subset-split, delete, reorder, and insert*, leads to an improvement in recommendation performance, highlighting the synergistic effect between these strategies. Particularly, on the Beauty and Sports datasets, the *slide-window + subset-split* achieves the highest performance among all augmentation combinations, with an average relative improvement of **7.2%** in Recall@20 and **8.3%** in NDCG@20 compared to using *slide-window* alone.

Contrastive learning methods also exhibit notable performance improvements when integrated with the *slide-window* strategy. For instance, on the Yelp dataset, the combination of CoSeRec and the *slide-window* strategy outperforms the use of CoSeRec alone, achieving increases of **84.4%** in Recall@20 and **66.3%** in NDCG@20. Similarly, on the ML-1m dataset, the combination of ICLRec and the *slide-window* strategy achieves the best performance in most cases, leading to improvements of **2.6x** in Recall@20 and **3.1x** in NDCG@20 compared to using ICLRec alone. These results indicate

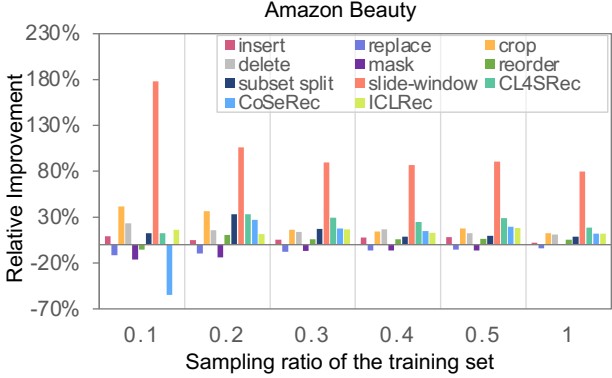

**Figure 3: Performance improvements (Recall@20) of each data augmentation strategy over backbone model (i.e. SAS-Rec) on Amazon Beauty dataset.**

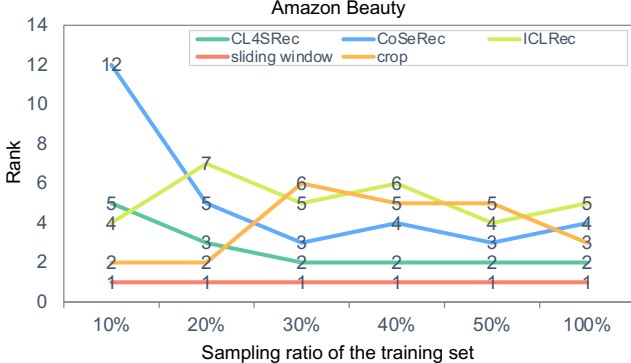

**Figure 4: Performance ranking variations of two data augmentation strategies and three contrastive learning methods in various cold-start scenarios.**

that integrating the *slide-window* strategy with various sequence-level augmentation techniques or contrastive learning-based SR methods can further boost recommendation performance. It is noteworthy that contrastive learning methods exhibit significant performance improvements over pure data augmentation methods only when employed in conjunction with the *slide-window* strategy. Otherwise, contrastive learning methods demonstrate comparable or even inferior performance compared to certain data augmentation strategies.

## 4.3 Cold-start Performance (RQ2)

To compare different data augmentation strategies and contrastive learning methods in the cold-start scenario, we first simulate different levels of cold-start by randomly sampling the training data at proportions of [0.1, 0.2, 0.3, 0.4, 0.5]. Then, we apply each data augmentation strategy or contrastive learning-based SR method to the sampled training set and train the models using the augmented data. Finally, we assess the performance of all models on the original test set. Fig. 3 illustrates the relative performance improvement ratios of the different methods compared to SASRec (No

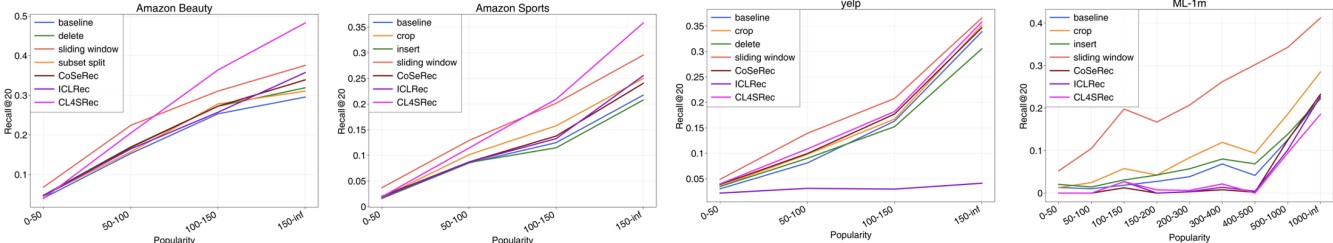

**Figure 5: Performance comparison of different data augmentation and contrastive learning methods under different item popularity. For each dataset, we select the top-performing three data augmentation methods for comparison. Baseline denotes no augmentation is utilized.**

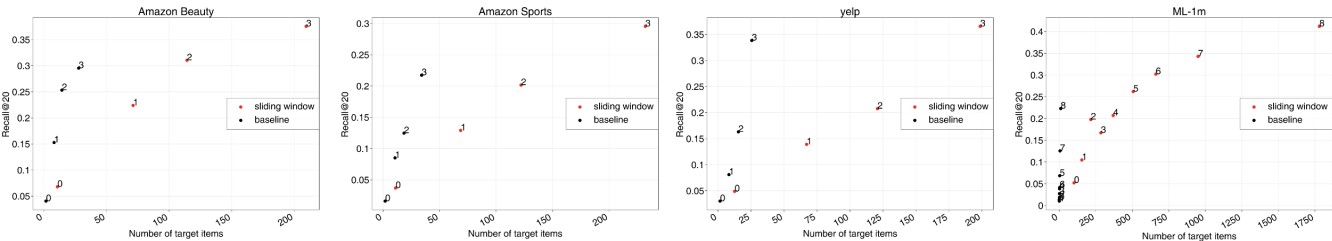

**Figure 6: Performance comparison of slide-window augmentation and baseline (no augmentation) w.r.t. the change of the popularity of target items. The x-axis denotes the average number of target item from test set as the target item in the training set. The text next to the scatters denote which popularity interval it belongs to. For example, for the Amazon Beauty dataset, "0" denotes popularity interval 0-50, "1" represents interval 50-100, and so on.**

augmentation) on Amazon Beauty dataset. We observe that in most cases, the less training data available, the more significant the relative performance improvement brought by different augmentation strategies.

Furthermore, we rank different contrastive learning methods and sequence-level augmentation methods (*slide-window* and *crop*) based on their performance (Recall@20) in various cold-start scenarios and depict the changes in their rankings in Fig 4. It is evident that *slide-window* consistently outperforms other sequence-level augmentation methods and contrastive learning methods across all five cold-start settings. Interestingly, while *crop* performs less favorably than contrastive learning methods on the original training set (i.e., sampling ratio equals 100%), it surpasses all contrastive learning methods when the sampling ratios are reduced to 10% and 20%.

## 4.4 Discussion on the Improved Performance (RQ3)

We compare the top-performing data augmentation methods and contrastive learning-based methods under varying item popularity on four benchmark datasets, using Recall@20 as the performance metric. In this work, the item popularity is defined as the frequency of each item as the target item in the training set. With such standard, the partition of sequences with different popularity is fixed and all data augmentation methods can be evaluated and fairly compared using the same test sequences. As shown in Fig 5, we can observe that all these methods, no matter direct data augmentation

or contrastive learning-based ones, tend to perform better when the target item has higher popularity. To further explore how simple data augmentation methods can have significant performance gains, we use the slide-window method as an example to test the effect of item popularity change on the model performance. In Fig 6, the ticks on the x-axis denote the average number of target item from test set as the target item in the training set. Addtionally, the text next to the scatters denotes which popularity interval it belongs to, and scatter with same number indicates they belong to the same target item popularity interval (e.g., scatters with text "0" in *Amazon Sports* denotes they come from the first popularity interval, namely 0-50, of *Amazon Sports* shown in Fig 5. As shown in Fig 6, the variation in model performance with the number of target items is approximately linear in the beginning, and then we can observe marginal utility of the performance improvement (in ML-1m dataset). This result, to some extent, indicates the model performance improvement brought by slide-window is obtaine by increasing the number of target item from test set as the target item in augmented training set.

## 4.5 In-depth Analysis (RQ4&5)

**Impact of the size of data augmentations *n*.** We compare the Recall@20 for different augmentation strategies when *n* is set to 2, 3, 5, and 10. Fig 7 presents the results on the Amazon Beauty dataset, where the left plot shows the results using a single sequence augmentation, and the right plot shows the results of slide-window augmentation combined with other augmentation strategies. We

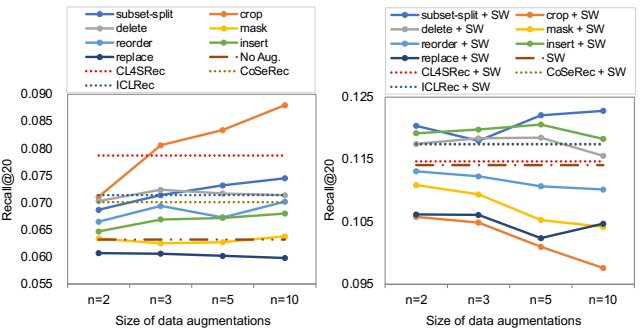

Figure 7: Impact of the size of data augmentations.

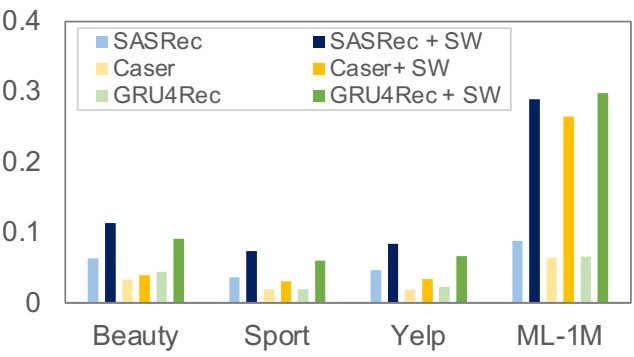

Figure 8: Effectiveness of slide-window on different model architectures.

observe that when using a single strategy, the Recall@20 for most augmentation strategies increases with the number of augmentations. However, when combined with slide-window augmentation, more than half of the augmentation strategies show a decrease in Recall@20 with larger size of augmentations. This could be attributed to the fact that excessive random augmentations introduce too much random noise during model training, resulting in the model's inability to accurately capture user interests.

**Impact of the sampling strategy.** Memory-based sampling strategies consistently outperform baseline and random-based sampling strategies, as demonstrated in Fig 10. We could observe that the performance gap between memory-based methods and random-based methods remains as the memory-based sampling could provide high-quality samples by utilizing correlation in memory rather than random. Also, the replacement strategy shows a fluctuating curve compared with the baseline while the insertion achieves stable enhancement. This is consistent with our observation in Tab 2.

**Impact of the slide-window size.** For slide-window augmentation, we also explore the impact of slide window size on recommendation performance. Due to space limitation, we only present the results on the beauty dataset. As shown in Fig. 9, we observe that different slide window sizes do not have a significant impact on recommendation performance, which may be due to the short average sequence length of the beauty dataset.

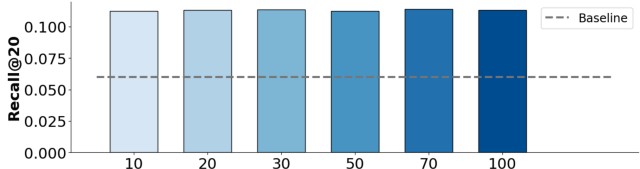

Figure 9: Effectiveness of sliding window size on Beauty.

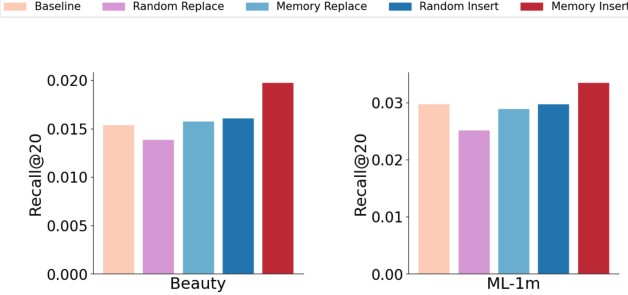

Figure 10: Comparison of augmentation sampling strategy.

**Whether different model architectures can benefit from the slide-window strategy?** We select three classic sequential recommendation models: SASRec[16], Caser[34], and GRU4Rec [10], representing transformer-based, CNN-based, and RNN-based SR models, respectively. For each model architecture, we conduct training using the original training set and the training set augmented with slide-window separately. The performance of these models is shown in Fig 8. We find that after augmentation with the *slide-window*, the performance of the three model architectures significantly improves on all four datasets, particularly on ML-1m, where slide-window provides a **3x** to **4x** performance improvement for the three models. This indicates that the slide-window sequence augmentation strategy is highly applicable and can provide more effective supervision signals for model training.

**Training and inference time.** As shown in Tab. 4, although the utilization of single or combined data augmentation strategies inevitably increases the volume of training data, their training time is still lower than contrastive learning methods. This is because contrastive learning methods usually introduce auxiliary tasks and complex positive view construction strategies, which increase computational overhead. Additionally, sequence-level augmentation strategies do not increase the model's inference time, as they only affect the training phase of the backbone model.

## 5 RELATED WORK

### 5.1 Sequential Recommendation

Intensive studies about recommender systems of various real-world scenarios found that sequential behaviors are important signals to model user preferences. And many efforts have been devoted to leveraging sequential behaviors to better capture behavior patterns. The very early and the most intuitive method is adopting the Markov Chain assumption for sequential recommendation [9, 29],

**Table 4: Comparison of training time and inference time between different data augmentation strategies and contrastive learning methods.**

| Methods | Alone | | +SW | | Test time (s) |
|---|---|---|---|---|---|
| | Recall@20 | Training time (s) | Recall@20 | Training time (s) | |
| No aug. | 0.065 | 335.76 | 0.114 | 691.72 | 0.20 |
| slide-window | 0.114 | 691.72 | - | - | - |
| subset-split | 0.070 | 463.32 | 0.120 | 1232.58 | 0.20 |
| replace | 0.062 | 495.05 | 0.103 | 1443.56 | 0.20 |
| reorder | 0.069 | 453.19 | 0.114 | 1766.25 | 0.19 |
| mask | 0.063 | 510.61 | 0.109 | 1064.83 | 0.20 |
| insert | 0.068 | 431.12 | 0.118 | 2023.55 | 0.20 |
| delete | 0.071 | 467.31 | 0.119 | 2642.98 | 0.19 |
| crop | 0.077 | 539.55 | 0.105 | 1894.35 | 0.19 |
| CoSeRec | 0.070 | 4650.78 | 0.117 | 13800.46 | 0.25 |
| CL4SRec | 0.079 | 3321.12 | 0.114 | 8941.15 | 0.20 |
| ICLRec | 0.071 | 1436.71 | 0.117 | 4055.04 | 0.26 |

where the next interaction is conditional on the past few interactions. Later, with the population of deep learning, many DL-based models were proposed to model sequential behaviors. GRU4Rec [10] is one of the most well-known SR models, of which Gated Recurrent Unit (GRU) is first introduced to model sequential behaviors. In addition, many other deep learning models were also introduced to seek better performance, such as Recurrent Neural Network (RNN) [21], Convolutional Neural Network (CNN) [35], Graph Neural Network (GNN) [2], and Multilayer Perceptron (MLP) [47]. Except for the aforementioned models, attention-based models are being intensively studied and are widely being adopted in sequential recommendation tasks [17, 32]. Besides, there are many interesting ongoing works focusing on other techniques like contrastive learning [4, 22, 38], reinforcement learning [40], and relation awareness [13].

## 5.2 Contrastive Learning for Recommendation

Contrastive Learning (CL) aims to improve the quality of representations by reducing the distance between positive views generated from the same data instance while separating them from negative views in a latent space. In the field of sequential recommendation, sequence-level data augmentation or feature-level data augmentation is often used to create positive views, with augmented views of other data instances in the same training batch serving as negative views. For instance, CL4SRec [38] employed three sequence-level data augmentation techniques, namely cropping, masking, and reordering, to construct positive views. Subsequently, CoSeRec [22] proposed to generate robust augmented sequences based on item correlations. To mitigate the representation degradation, DuoRec [28] utilized feature-level augmentation based on dropout to better maintain semantic consistency between positive views. Despite these methods claiming that contrastive learning can significantly enhance the performance of recommender systems, they do not consider direct data augmentation as a baseline and thus cannot ascertain whether contrastive learning has a distinct advantage in mitigating data sparsity compared direct data augmentation.

## 5.3 Data Augmentation for Recommendation

Data augmentation is an effective method to improve the performance of DL-based models, particularly when the training data

is scarce. In CV and NLP, data augmentation has drawn much attention and is widely adopted in model training. However, as for recommender systems, compared with CV or NLP, studies regarding data augmentation are still at a rather rudimentary stage.

For sequential recommendation tasks, basic data augmentation approaches create augmented sequences out of the original sequences themselves through simple transformations (*e.g.,* crop, reorder), small perbulation (*e.g.,* noise/redundancy injection, synonym replacement [30]), or subset selection (*e.g.,* slide-window [35], subset split [33], and item masking [30]). Recent works regarding the aforementioned approaches focus on time-aware approaches, which better retain time coherence between the augmented sequences and the original ones, and further enhance the model's performance [6, 27]. Except for the aforementioned basic approaches, some data augmentation approaches choose to create highly plausible sequences by synthesizing and injecting/prepending fake samples into the original sequence [11, 14, 23], or modeling counterfactual data distribution [37, 44]. Apart from being applied in the sequential recommendation, data augmentation techniques are also applied in collaborative filtering to alleviate the data sparsity problem [36] or bypass negative sampling [19] during the model training.

The work most related to ours is [30], which explored the impact of four augmentation strategies on sequential recommendation, namely noise injection, redundancy injection, item masking, and synonym replacement. Different from it, our work benchmarked various direct data augmentation methods and contrastive learning methods, providing comprehensive analysis of the effectiveness of sequence-level data augmentation in sequential recommendation research, and offers insights into the improvements achieved.

## 6 CONCLUSION AND FUTURE WORK

In this paper, we benchmark eight widely used sequence-level data augmentation strategies, as well as three state-of-the-art contrastive learning SR methods, on four real datasets under both full data and cold-start settings. The results reveal that the performance of SRS can be improved by solely relying on data augmentation without using contrastive learning paradigm. Therefore, the current research community might underestimate the effectiveness of simple sequence-level augmentation and excessively emphasize the necessity of contrastive learning for sequential recommendation tasks. In the future, we will extend the scope of benchmarking to include a broader range of data augmentation strategies and contrastive learning methods, providing theoretical justification for the effectiveness of sequence augmentation.

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
