# OpenReview forum: "Is Contrastive Learning Necessary? A Study of Data Augmentation vs Contrastive Learning in Sequential Recommendation"
_ACM.org/TheWebConf/2024/Conference — TheWebConf24_

### Official Review · Reviewer_UeyS · 2023-11-23

**Novelty:** 5
**Technical Quality:** 5

**Review:**

The paper, through meticulous experimentation, demonstrates that optimal performance can be achieved through simple data augmentation or the combination of various data augmentation methods. The results even surpass some current CL-based methods, suggesting that there may be no need to design complex contrastive losses in the future.
These findings have the potential to simplify the development of temporal recommendation, guiding researchers towards more fundamental studies in contrastive learning. The emphasis should shift from attempting to increase the complexity of contrastive learning to enhancing recommendation effectiveness.


----------- Strengths -----------
1. The authors hope to inspire researchers to conduct more fundamental studies on the key functional components of CL techniques rather than finding ways to come up with more and more complex comparative learning methods.
2. The authors demonstrate with exhaustive experiments that the effectiveness of sequence recommendation can be improved by data augmentation.
3. The paper is well-organized and easy to follow.

----------- Weaknesses -----------
1. The authors say that the data augmentation strategy is simpler and more effective compared to CL-based methods, and the experimental section proves the effectiveness through a large number of experiments but lacks the associated complexity analysis.
2. I noticed that in the experimental section, the authors give two baseline models, CL4SRec and ICLRec, on the Beauty and Sports datasets, which are quite different from those given in the original article.
3. In Figure 9, what you're trying to say is that different slide window sizes don't have a significant effect on recommendation performance, but there should be some differences between them, even if they're small.

**Questions:**

Please see the weakness part.

**Reviewer Confidence:**

3: The reviewer is confident but not certain that the evaluation is correct

**Scope:**

3: The work is somewhat relevant to the Web and to the track, and is of narrow interest to a sub-community

---

### Official Review · Reviewer_o33W · 2023-11-29

**Novelty:** 4
**Technical Quality:** 5

**Review:**

**Summary**

The paper presents an empirical study on the role of contrastive learning (CL) in data augmentation-based sequential recommendation systems (SRS). This seems like an unexplored and understudied problem where a thorough empirical investigation may benefit the research community. The paper poses and investigates the research question – whether CL is necessary to improve SRS performance. Despite the growing popularity of CL in SRS, the paper makes an intriguing conclusion -- data augmentation is sufficient, and CL is not necessary.



**Pros**

- This paper appears to be the first comprehensive study that compares 8 different augmentation strategies and 3 popular CL baselines. It also studies the effect of augmentation size and sampling strategy.

- The research questions are mostly meaningful and impactful. The experiments are carefully designed to answer these questions.



**Cons**

- *Questionable results*: The results of CL baselines in Table 2 are wildly different from the reported ones in their respective original papers.

- *Weird results in RQ4*: The sequence length in the beauty dataset is <10 per user. However, according to the results presented in Fig. 7, it appears that using up to 10 augmentations (which may include deletions) in a sequence does not affect recall.

- *Unclear insights in RQ3*: The authors conclude from Fig. 5 that all methods perform well for popular items. It is not clear why this is an insightful observation. Also, at least from this figure, it is not clear if there is a clear winner between data augmentation and CL methods across different item groups.

- *Low Reproducibility*: Code not provided.



**Recommendation**

This is a borderline paper due to questionable confidence in experimental results and conclusions drawn from them. As an empirical study, the lack of confidence or robust justification undermines its contribution.

**Questions:**

- Please justify the discrepancy in the results of baseline CL methods (see cons).

- Would analyzing performance across different user groups, rather than item groups (RQ3), provide more meaningful insights? It is suggested that such analyses be added to the paper.

- Are the results (recall and ndcg) reported as percentages? This is not explicitly mentioned.

- The different colors in Fig. 5 are very hard to distinguish.

**Ethics Review Description:**

no ethical concerns

**Reviewer Confidence:**

3: The reviewer is confident but not certain that the evaluation is correct

**Scope:**

4: The work is relevant to the Web and to the track, and is of broad interest to the community

---

### Official Review · Reviewer_ch1L · 2023-11-30

**Novelty:** 4
**Technical Quality:** 5

**Review:**

This work presents a critical comparison of combining data augmentation or contrastive learning (CL) for the sequential recommender systems (SRS), which often suffers from data sparsity. Experiments on four datasets, comparing different data augmentation strategies with state-of-the-art CL-based SRS methods, showing that data augmentation is sufficient and CL may not be necessarily required.

**Pros:**
- (P1) The work systematically studies different data augmentation techniques for sequence data and how they contribute to the sequential recommendation task compared to some state-of-the-art CL-based methods. Notably, slide-window strategy, an augmentation technique, can significantly enhance the performance of SRS (including CL-based methods) and has a synergistic effect when combined with other data augmentation techniques.
- (P2) The codes and datasets of this work will be provided for reproducibility.
- (P3) The experimental results look promising and show relative improvements over baselines.

**Cons:**
- (C1) Slide-window (SW) strategy for sequential data augmentation produces rich augmented data that also preserves sequential context. This obviously significantly increases the amount of data for training. This augmentation strategy thus enhances the performance in contrast to training without SW. Moreover, SW also helps to enhance other CL-based methods performance. The results in Table 3 show that SW also enhances other CL-based methods to achieve the best ranking performance on Yelp and ML-1m datasets. It is not clear how SW combined with other augmentation strategy might superior CL-based methods with SW.
- (C2) On Table 3, it creates an unfair advantage when comparing CL-based + SW with SASRec + SW + another augmentation strategy. It is good to show whether applying similar augmentation strategies when comparing CL-based with non-CL-based models to show the advantages or disadvantage when using CL-based models over non-CL-based models. For example, comparing SASRec + SW + subset-split and ICLRec + SW + subset-split.

**Questions:**

I have only one question related to (C2) discussed above.
- (Q1) How do CL-based models perform when applying SW + another augmentation strategy (if possible)? This will help clarify whether applying CL with the same augmentation strategies enhances ranking performance.

**Reviewer Confidence:**

3: The reviewer is confident but not certain that the evaluation is correct

**Scope:**

4: The work is relevant to the Web and to the track, and is of broad interest to the community

---

### Official Review · Reviewer_pDy8 · 2023-11-30

**Novelty:** 2
**Technical Quality:** 2

**Review:**

This paper designs a comparison framework to evaluate two approaches in sequential recommendations, Data Augmentation and Contrastive Learning. The intellectual contributions of this paper are weak in my opinion for a conference at WWW level. I found the paper to be focused too much on application, seems to be a benchmarking paper. Additionally, the contributions to web community are not clearly stated which is required by the conference.

**Questions:**

Major:

1- All evaluated datasets are quite sparse. Experimenting also on a slightly denser dataset will be nice for comparison.

2- How could hyperparameter optimization for different methods be carried out? In the text, it says that 0.5 dropout has been used for all transformers. However, that will not be fair.

3- The contribution to the web community is not clearly stated anywhere in text, which is required by the conference.

Minor:

1- Domain specific evaluation metrics should be referenced, and explained in the text.

2- Ablation study should be performed more in depth.

**Reviewer Confidence:**

4: The reviewer is certain that the evaluation is correct and very familiar with the relevant literature

**Scope:**

2: The connection to the Web is incidental, e.g., use of Web data or API

---

### Official Review · Reviewer_cchU · 2023-12-06

**Novelty:** 4
**Technical Quality:** 4

**Review:**

This work investigates the role of data augmentation and contrastive learning in sequential recommendation systems (SRS). The authors conduct a comprehensive experimental study comparing the performance of SRS based on data augmentation alone and those based on contrastive learning. They explore the effectiveness of eight popular sequence-level augmentation strategies and benchmark them against state-of-the-art contrastive learning-based methods on real-world datasets. The results show that certain data augmentation strategies can achieve comparable or even superior performance to contrastive learning-based methods while requiring less training and inference time. This study highlights the potential of data augmentation as a standalone technique to address the data sparsity issue in SRS and questions the necessity of contrastive learning in this context.

The findings of the study validate the efficacy of data augmentation as a standalone technique for improving the performance of SRS. The results demonstrate that certain sequence-level data augmentation strategies can effectively mitigate the data sparsity issue, achieving comparable or even superior performance compared to contrastive learning-based methods.

To improve this work, the authors could consider including detailed parameter tuning and settings of the compared baselines in the evaluation section. This additional information would provide a clearer understanding of the experimental setup and allow for better reproducibility of the results. Specifically, the authors could provide details on the hyperparameters and configurations used for each baseline method, including the contrastive learning methods and data augmentation strategies.

To enhance the comprehensiveness of the work, conducting a more detailed experiment comparison between the newly proposed method and baseline models in terms of computational and memory cost would be beneficial. This additional analysis would provide insights into the efficiency and resource requirements of the different approaches, which is an important aspect to consider in real-world applications. The authors could compare the training and inference times of each method, measure the memory usage during the experiments, and report any significant differences observed. Additionally, they could explore the scalability of the proposed method and baselines by varying the dataset size or model complexity to assess their performance under different computational and memory constraints.

**Questions:**

To improve this work, the authors should include detailed parameter tuning and baseline settings in the evaluation section. This would enhance the understanding and reproducibility of the experimental setup. Additionally, conducting a more comprehensive experiment comparison between the proposed method and baselines in terms of computational and memory cost is necessary. This analysis would provide insights into the efficiency and resource requirements of each approach, benefiting real-world applications. Comparing training and inference times, measuring memory usage, and exploring scalability would contribute to a more comprehensive evaluation.

**Reviewer Confidence:**

3: The reviewer is confident but not certain that the evaluation is correct

**Scope:**

3: The work is somewhat relevant to the Web and to the track, and is of narrow interest to a sub-community

---

### Decision · Program_Chairs · 2024-01-22

**Decision:**

Accept

**Comment:**

If we're willing to treat the one negative review as an outlier, the remaining reviews make a pretty good argument for acceptance. The rebuttal is fairly persuasive (though not all reviewers engaged with it), and most of the issues raised seem to be about issues that could be clarified in a revision (though there are quite a lot of these, and I dare say the authors have a lot to do to revise the paper).